# Astrocytic Neuroimmunological Roles Interacting with Microglial Cells in Neurodegenerative Diseases

**DOI:** 10.3390/ijms24021599

**Published:** 2023-01-13

**Authors:** Mari Gotoh, Yasunori Miyamoto, Hiroko Ikeshima-Kataoka

**Affiliations:** 1Department of Clinical Laboratory Medicine, Faculty of Medical Technology, Teikyo University, 2-11-1, Itabashi-ku, Tokyo 173-8605, Japan; 2Institute for Human Life Science, Ochanomizu University, 2-1-1 Ohtsuka, Bunkyo-ku, Tokyo 112-8610, Japan; 3Department of Biology, Keio University, 4-1-1, Hiyoshi, Kohoku-ku, Yokohama 223-8521, Japan; 4Department of Biosciences and Informatics, Faculty of Science and Technology, Keio University, 3-14-1 Hiyoshi, Kohoku-ku, Yokohama 223-8522, Japan; 5Faculty of Science and Engineering, Waseda University, 3-4-1 Okubo, Shinjuku-ku, Tokyo 169-8555, Japan

**Keywords:** astrocyte, blood-brain barrier, 2ccPA, cPA, extracellular matrix, microglia, neurodegenerative disease, neuroinflammation, reactive astrocytes, reactive microglia, TN-C, traumatic brain injury

## Abstract

Both astrocytic and microglial functions have been extensively investigated in healthy subjects and neurodegenerative diseases. For astrocytes, not only various sub-types were identified but phagocytic activity was also clarified recently and is making dramatic progress. In this review paper, we mostly focus on the functional role of astrocytes in the extracellular matrix and on interactions between reactive astrocytes and reactive microglia in normal states and in neurodegenerative diseases, because the authors feel it is necessary to elucidate the mechanisms among activated glial cells in the pathology of neurological diseases in order to pave the way for drug discovery. Finally, we will review cyclic phosphatidic acid (cPA), a naturally occurring phospholipid mediator that induces a variety of biological activities in the brain both in vivo and in vitro. We propose that cPA may serve as a novel therapeutic molecule for the treatment of brain injury and neuroinflammation.

## 1. Introduction

### 1.1. What Are Astrocytes?

Astrocytes are the most prevalent glial cells in the brain and play a variety of roles in the healthy brain, including in neurogenesis, immune functions, blood-brain barrier (BBB) integrity, and learning and memory. Heterogeneous astrocytic functions, morphological, developmental, at the molecular level, physiological, and functional, have recently been reported both in healthy states and central nervous system (CNS) diseases [1]. Anatomically, astrocytes have been shown to possess layer-specific properties between neurons in the mouse somatosensory cortex [2]. Online databases for transcriptomes and proteomes of astrocytes have been concisely summarized in a review paper [3]. 

### 1.2. Astrocytes in the Extracellular Matrix

Astrocyte endfeet attach to the basement membrane of endothelial cells and constitute the BBB integrity. In the injured brain, astrocytes regulate the degradation of the basement membrane and modulate the integrity and permeability of the BBB. In addition, activated astrocytes upregulate the secretion of tenascin-C (TN-C), and TN-C regulates the expression of inflammatory cytokines and neuron survival. We summarize astrocytic function in the extracellular matrix and in relation to TN-C in detail in Section 2.

### 1.3. Reactive Astrocytes

Astrocytes are known to become active after brain injury, after inflammation develops in the brain, in neurological diseases, and during aging [4]. Reactive astrocytes include both neurotoxic astrocytes and neuroprotective astrocytes, termed A1 astrocytes and A2 astrocytes, respectively [5]. The authors characterized subsets of reactive astrocytes by using GeneChip arrays, and their phenotypes are strongly dependent on the type of brain damage induced. The authors also demonstrated differences between the levels of expression of extracellular binding/adhesion/modification genes, cytokine signaling genes, antigen presentation, and complement pathways in middle cerebral artery occlusion (MCAO) reactive astrocytes and lipopolysaccharide (LPS) reactive astrocytes. Furthermore, their colleagues termed these reactive astrocyte subsets A1 astrocytes and A2 astrocytes depending on whether they were neuroinflammation-induced neurotoxic astrocytes or ischemia-induced neuroprotective astrocytes [6,7]. The authors subsequently demonstrated that the neurotoxic A1 astrocytes were induced by activated microglia in response to secretion of interleukin (IL)-1α (IL-1α), tumor necrosis factor (TNF)α, and complement component subunit 1q (C1q).

The state of astrocyte activation caused by traumatic injury, neurodegenerative diseases, or infection in the brain is referred to by many names, including “astrocyte reactivation”, “reactive gliosis”, and “astrocyte reaction”. The nomenclature and definitions have recently been summarized in detail by experts in the field of reactive astrocytes [8]. Considering reactive astrocyte-targeting therapy could succeed in developing a treatment for neurodegenerative and neuroinfectious disorders. 

#### 1.3.1. Reactive Astrocytes in Neurological Diseases

CD49f, encoded by the integrin alpha 6 gene, has been reported to be a novel marker for human induced pluripotent stem cell (hiPSC)-derived A1 astrocytes, which are toxic to neurons in vitro, respond to inflammatory stimulation with TNF-α, IL-1α, and C1q while maintaining its expression [9]. The sorting strategy for CD49f+ cells from heterogeneous hiPSC-derived cultures enriched mature astrocytes. Furthermore, it has been possible to isolate CD49f+ astrocytes from human fetal brains; they are also present in human adult brains and maintain physiological astrocytic functions. However, human A1-like astrocytes lack phagocytic activity and glutamate uptake. The authors concluded that CD49f+ is a reactivity-independent astrocyte marker and may provide a powerful tool for investigating molecular mechanisms in A1-astrocyte-dependent brain diseases.

Optic nerve crush induces the release of toxins from reactive astrocytes that have killed damaged retinal ganglion cells (RGCs) in a glaucoma model. Namely, the damage is required for neurons to become susceptible to astrocyte toxin-induced death. The authors claimed that a second signal, such as injury, is needed for the death of RGCs [10].

Under normal conditions, one of the water channels, aquaporin 4 (AQP4), is expressed in only the endfeet of astrocytes, but its localization changes to the whole cell body of reactive astrocytes in pathological states [11]. The lack of AQP4 in the retina resulted in the loss of retinal ganglion cells in MOG (35-55)-induced experimental autoimmune encephalo-myelitis (EAE) and in impairment of the blood–retina barrier in autoimmune CNS inflammation [12]. Thus, AQP4 expression in Muller cells, which are related to astrocytes, might be important for water drainage during the development of the inflammatory response. 

#### 1.3.2. Reactive Astrocytes in Blood-Brain Barrier (BBB)

We investigated the functional role of reactive astrocytes in mice with a stab wound injury to the cerebral cortex as a traumatic brain injury (TBI) model and in primary astrocyte cultures exposed to LPS as a neuroinflammation model, and the results revealed a close relationship between astrocyte reactivation and recovery from BBB breakdown caused by TBI or inflammation [13,14,15]. 

We have confirmed that astrocytic endfeet do not play a major role in BBB integrity by using an in vivo imaging analysis system in a study of GFP-driven eGFP mice [16]. Astrocyte reactivation in response to brain injury or infection leads to the retraction of their endfeet from blood vessels, and that increases the permeability of the BBB [17,18]. Thus, although astrocytic endfeet are not directly involved in the physical barrier of the BBB, molecules secreted from astrocytic endfeet may tighten the tight junction such as Claudin5 (Cldn5), occluding, ZO-1, and -2 between endothelial cells at the BBB in the brain.

## 2. Astrocytes and the Extracellular Matrix

### 2.1. Extracellular Matrix in the Brain 

In brain tissue, the extracellular matrix in the brain parenchyma and basement membrane contributes to homeostasis [19]. The basement membrane is an important component of the BBB and plays an important role in the integrity and permeability of the BBB [20,21,22], and endothelial cells, pericytes, and astrocytes are attached to the basement membrane [20,21]. The basement membrane is composed of four major glycoprotein families: the laminin (LN) family, collagen type IV family, nidogen family, and heparan sulfate proteoglycan (HSPG) family [19]. Two members of the HSPG family, perlecan and agrin, are major components of the basement membrane of the BBB. The brain parenchyma contains the chondroitin sulfate/dermatan sulfate proteoglycan family (biglycan, decorin, neuron-glial antigen 2 [NG2], aggrecan, brevican, neurocan, versican, brevican, etc.), tenascin C (TN-C), tenascin R, reelin, etc. [19].

### 2.2. Astrocytes and the Basement Membrane

Astrocyte endfeet attach to the basement membrane and surround brain vascular vessels. The endfeet of astrocytes are firmly attached to the basement membrane and together they constitute the BBB. The dystroglycan (DG) on the endfeet of the astrocytes is bound to laminin (LN) and agrin in the basement membrane. DG binds to the actin cytoskeleton through the dystrophin-associated protein (DAP) complex and dystrophin [21], and the DAP complex is composed of syntrophin (Syn) and dytrobrevin (DB) [23]. The water channel protein AQP4 and a potassium channel, Kir4.1, bind to dystrophin via Syn [21,24,25,26]. In addition, the gap junctions between the astrocyte endfeet contribute to the organization of AQP4. The gap junctions are composed of connexin 43 (Cx43) and Cx30 [27]. These connections also affect the integrity and permeability of the BBB [23]. The localization of AQP4 on the endfeet of astrocytes is responsible for the flow of water from the blood vessels into the brain parenchyma. This flow is called the glymphatic system [28]. The glymphatic system has been suggested to be responsible for clearing waste products and pathogenic substances from the brain parenchyma, and it also contributes to the clearance of amyloid β40 [29], suggesting that the disruption of BBB integrity contributes to the onset of Alzheimer’s disease (AD).

The production of matrix metalloproteinase 9 (MMP-9) and MMP-2 by invading neutrophils and MMP-9 by pericytes increases during the inflammation that develops after an injury [30]. The MMPs degrade agrin and dystroglycan and destroy the integrity of the BBB [31,32], and its destruction disrupts the localization of AQP4 on the sites of attachment of the endfeet. The loss of AQP4 localization prevents water flow through AQP4 and results in edema [33]. Activated astrocytes were induced to express antichymotrypsin and thereby cause BBB disruption in a human-induced pluripotent stem cell (iPSC)-derived BBB co-culture model [34]. However, MMP-3 expression by activated astrocytes is suppressed following cerebral ischemia, and the downregulation of MMP-3 contributes to the suppression of BBB breakdown [35]. The function of activated astrocytes remains unclear.

### 2.3. Tenascin-C and Astrocytes

Changes in the levels of several extracellular matrix proteins, TN-C, and reelin in the brain have been reported in the acute phase of an experimental rat model of TBI [36] and the modulation of the extracellular matrix is a potential target for the treatment of TBI [37]. Reactive astrocytes are known to express extremely high levels of TN-C when the brain is damaged [38,39,40]. Cerebrospinal fluid TN-C concentrations have been shown to be positively correlated with subarachnoid hemorrhage (SAH) severity [41] and increases in serum and cerebrospinal fluid TN-C protein levels are closely associated with trauma severity and clinical outcomes after TBI [42,43].

The upregulated TN-C in TBI regulates inflammation and neuronal protection during neurodegeneration. Liu et al. investigated the neurotoxic function of TN-C and found that TN-C deficiency protects against neuronal cell death in a mouse model of SAH [44]. Silencing of TN-C has been shown to inhibit inflammation and apoptosis via the PI-3 K/Akt/NF-kB signaling pathway in an SAH cell model [45], and TN-C deficiency was found to promote anti-inflammatory cytokine expression in an autoimmune glaucoma mouse model [46]. These findings indicate that TN-C can function as a neurotoxic factor in the brain. Research on the neuroprotective function of TN-C has revealed that α9 integrin promotes neurite outgrowth in the injured CNS, where TN-C is upregulated [47], and that the interaction between TN-C and αvβ3 integrin promotes cell proliferation and inhibits apoptosis [48]. The addition of extracellular vesicles from interleukin-1β-treated TN-C cultured astrocytes has been observed to increase wound recovery in a scratch model [49], and TN-C upregulation induced by a metabolically stabilized derivative of cyclic phosphatidic acid, 2-carba cyclic phosphatidic acid (2ccPA), has been shown to contribute to the survival of neurons in the vicinity of stab-wounds [40] (we explain cPA in detail in Section 7 below). These findings support the notion that TN-C has dual functions with respect to neuroprotection and wound recovery. The difference between these TN-C functions may be attributable to differences between splice variants of TN-C. The injury-induced splice variant of TN-C may influence axonal regeneration and repair processes in the damaged CNS [50]. TN-C modulates the balance between neuroprotective and neurotoxic inflammatory cytokine expression in the injured brain [14].

## 3. What Are Microglia?

### 3.1. Development of Microglia

Microglia have been reported to be derived from erythro-myeloid progenitors (EMPs) from the yolk sac in the blood island [51]. EMPs differentiate in a PU.1- and IRF8-dependent manner and infiltrate the brain from the embryonic circulation [52]. During normal brain development, microglia function in synapse formation and pruning [53,54,55], in neuronal activity [56], and in the phagocytosis and removal of foreign objects and dead cells [57].

### 3.2. Reactive Microglia in Neurodegenerative Diseases

Microglia are related to the pathogenesis of most neurodegenerative disorders, including AD, Parkinson’s disease (PD), amyotrophic lateral sclerosis (ALS), MS, brain tumors, ischemia, mental illness, and neuropathic pain [58,59]. Researchers have recently published databases of the human microglial transcriptome in various brain regions, aging, and brain pathologies [60]. Microglia change their morphology from ramified to ameboid during epileptic seizures, and higher expression levels of fractalkine (CX3CR1) and ATP purine receptors (P2X4, P2Y4, P2Y6, P2Y12) in microglia have been shown to result in excessive neurogenesis or ectopic neural circuit formation [61].

Microglia switch from their resting state to an activated state depending on the status and environment of the brain. Researchers have recently clearly described microglial nomenclature [62].

### 3.3. Reactive Microglia in Neuropathic Pain

Tsuda et al. have reported finding that an extracellular ATP non-selective cationic channel, the P2X4 receptor, is specifically expressed in microglia in the dorsal horn of the spinal cord after nerve injury and that the functional inhibition and suppression of the expression of the P2X4 receptor controlled neuropathic pain [63,64,65,66]. They also found that CD11c+ microglia that had converted from resident microglia in response to nerve injury and expressed insulin-like growth factor 1 (IGF1) had the potential to relieve neuropathic pain [67]. In addition, the depletion of the CD11c+ microglia or prevention of IGF1 signaling was followed by a relapse of pain hypersensitivity in mice that had recovered. Targeting the CD11c+ microglia subset might lead to the development of drugs to treat neuropathic pain.

### 3.4. Reactive Microglia in Aging

Neuronal CD22 expression is known to inhibit microglial pro-inflammatory cytokine production [68], and the involvement of CD22 in aged microglia has recently been reported. Pluvinage et al. screened age-related modifiers of microglial phagocytosis and found that the canonically expressed B-cell recbptor CD22 is a negative regulator both in vivo and in vitro and is upregulated in aged microglia [69]. Blockade of CD22 has been demonstrated to promote clearance of myelin debris, amyloid-β oligomers, and a-synuclein fibrils and improve cognitive function in aged mice. The authors of that study claimed that murine models of ALS and Nieman-Pick type C as well as aged mice exhibited the upregulation of CD22 [70,71]. In short, CD22 may be a biomarker of aged microglia and may be suitable as a therapeutic target for aging.

The expression of the small GTPase Rhoa in adult microglia is important for normal brain function. Microglia that lack only the small GTPase Rhoa have been found to be neurotoxic and lead to AD-like pathology, LTP impairment, synapse loss, and memory deficits in mice [72]. The authors also showed that the Rhoa/Src signaling pathway was disrupted in the microglia of APP/PS1 AD model mice.

## 4. Phagocytic Activity of Reactive Astrocytes and Their Signaling

Astrocytes, as well as microglia, express phagocytotic activity on neuronal debris in damaged tissue. In the following section, we review the phagocytotic activity of reactive astrocytes in various CNS diseases.

An in vitro analysis performed in an adult mouse astrocyte culture by Wyss-Coray et al. revealed that astrocytes accumulated around amyloid β (Aβ) deposits and degraded them in response to monocyte chemoattractant protein-1 (MCP-1) [73]. In addition, the enhancement of lysosomal biogenesis in primary astrocytes by transcription factor EB (TFEB), a master regulator of lysosome activity, has been found to induce the uptake, trafficking, and degradation of Aβ [74]. Exogenous TFEB localization in astrocytes has been shown to enhance lysosomal function and reduce Aβ levels and amyloid plaque load in the hippocampus of amyloid precursor protein/presenilin-1 (APP/PS1) transgenic mice, a murine model of AD. Based on these observations, astrocytes have been concluded to provide adequate Aβ removal activity during the pathogenesis of AD, and astrocyte reactivity during the pathogenesis of AD has been reported [75]. Moreover, surprisingly, reactive astrocytes, not microglia, have been shown to contribute to the engulfment and digestion of presynaptic dystrophies in the hippocampus of APP/PS1 mice [76].

Astrocytic phagocytosis has been observed in myelin-injured patients, including in patients with multiple sclerosis (MS), progressive multifocal leukoencephalopathy, metachromatic leukodystrophy, and subacute infarcts, as well as in AD pathology [77]. Myelin debris has been observed in most of the hypertrophic astrocytes at sites of acute myelin breakdown. Moreover, it occurred through receptor-mediated endocytosis, resulted in NF-κB localization in myelin-laden astrocytic nuclei, and then led to chemokine secretion that recruited immune cells to the brain. Thus, astrocyte phagocytic activity on myelin is common in a variety of CNS pathologies. The discovery of the molecular mechanism of the regulation of astrocytic phagocytosis might provide an effective target for the treatment of CNS diseases. Astrocytic phagocytic activity after transient ischemic injury has also been reported in MCAO mice [78]. The phagocytic astrocytes were observed within the ischemic penumbra region, whereas phagocytic microglia were seen in the ischemic core region. Moreover, the upregulation of ABCA1 and its pathway molecules has been found to be essential for astrocytic phagocytosis both in vitro and in vivo. The authors proposed that further studies are needed to better understand the difference between the phagocytic activity of astrocytes and microglia.

Thus, targeting reactive astrocytes, which have heterogeneity and plasticity, for CNS diseases such as ischemic stroke, traumatic brain injury, or CNS demyelination should be elucidated [79].

## 5. Interaction between Reactive Astrocytes and Reactive Microglia

The crosstalk between astrocytes and microglia has been intensively investigated in the brain development and neurodegenerative diseases [80]. We have proposed that interactions between reactive astrocytes and reactive microglia in TBI in mice and in vitro in LPS-challenged astrocytes in primary cultures may be induced by osteopontin (OPN), one of the pro-inflammatory cytokine inducers [11,81,82]. Since receptors for OPN integrin α9β1 are expressed both in reactive astrocytes and reactive microglia, OPN signaling may be a key factor in the interactions between glial cells. On the other hand, the anti-inflammatory cytokines, IL-4, IL-10, IL-13, and TGF-β have been reported as potential key molecules between reactive astrocytes and activated microglia [83]. Moreover, Garland et al. reported a communication pathway between astrocytes and microglia in human brain diseases [84]. In the following section, we summarize recent findings regarding interactions among reactive glial cells in neurological diseases focusing on certain key factors.

### 5.1. Interactions between Reactive Glial Cells in AD

Hematopoiesis regulatory cytokine IL-3 production has been found to be extremely increased in AD and PD patients in comparison with healthy controls [85], and Ray et al. have reported that IL-3 is not only a factor that can be used to identify AD patients but it could also be the prediction of progression of the disease [86]. The source and target and the mechanism of IL-3 signaling in the pathology of AD have recently been clearly demonstrated by McAlpine et al. using IL-3 knockout mice crossing with 5X FAD mice, and tissue from the frontal cortex of AD patients [87]. They identified astrocytes as the source of IL-3 and that received by IL-3 receptor α (IL-3Rα) expressing microglia with countering Aβ deposits. In short, the above findings demonstrate that IL-3 is a mediator of the interactions between reactive astrocytes and microglia. In addition, the injection of recombinant IL-3 into the lateral ventricle of the brains of 5X FAD mice has been found to restore their memory. Thus, these findings show that IL-3 is a potential therapeutic target for the treatment of AD pathology.

Vandenbark et al. clearly summarized the interactions between astrocytes and microglia in AD and brain tumors in a review paper [88]. They stressed that the upregulation of apolipoprotein E (APOE) and triggering receptor expressed on myeloid cells 2 (TREM2) are correlated with interactions between astrocytes and microglia, in late-onset AD.

### 5.2. Interactions between Reactive Glial Cells in PD

NLY01, a potent glucagon-like peptide-1 receptor (GLP1R) agonist, has been found to block A1 neurotoxic astrocyte conversion by microglia, and it was shown to prolong the survival of dopaminergic neurons and reduce the pathology of hA53T α-synuclein transgenic mice, a PD model [89]. In addition, the repeated subcutaneous administration of NLY01 has been reported to block microglia-mediated reactive astrocyte conversion and the preservation of neuronal viability, resulting in the amelioration of spatial learning and memory in both 5XFAD and 3Xtg-AD mice [90]. Since NLY01 may have a neuroprotective role, it might be possible to use it to treat a variety of neurodegenerative diseases

### 5.3. Interactions between Reactive Glial Cells in ALS

Liddelow et al. found that activated microglia release IL-1α, TNFα, and C1q induce neurotoxic astrocytes [6], and they showed that knocking out these three factors extended the lifespan of SOD1^G93A^ ALS model mice [91]. Thus, regulating inflammatory cytokine production in reactive astrocytes precedes their neuroprotective role and may have the ability to cure ALS patients.

### 5.4. Interactions between Reactive Glial Cells in the BBB

Haruwaka et al. recently demonstrated a dual function of microglia in BBB integrity. At the early systemic inflammatory stage with an intraperitoneal injection of LPS, microglia contacted to chemokine CCL5 releasing endothelial cells and expressed Cldn5 to tighten BBB in a neuro-protective manner. However, during the prolonged inflammation in the late inflammatory stage induced by continuous LPS administration, the microglia phagocytosed astrocytic endfeet on blood vessels, which increased BBB permeability [92]. These findings revealed a novel function of microglia in which they contribute to BBB integrity by upregulating their expression of Cldn5. The same investigators also found Cldn5 expression in the microglial subset from the database, which was developed by deep single-cell RNA sequencing of microglia from various regions of various time points of developing mouse brains [93] (see also https://myeloidsc.appspot.com/, accessed on 16 January 2019).

### 5.5. Interactions between Reactive Glial Cells in Inflammation

IL-10 is known to function as an anti-inflammatory factor, and it has been found to have a neuroprotective function in vitro in microglia cultures [94], in a TBI mouse model [14], and in a murine stroke model [95]. Activated microglia and macrophages produce IL-10 [96]. An experiment conducted by co-culturing a primary culture of astrocytes and microglia revealed that IL-10 stimulates the production of transforming growth factor β (TGFβ) by astrocytes, and that the TGFβ activated the microglia [97]. These findings were confirmed by an experiment showing that the absence of IL-10 receptor/TGFβ signaling within astrocytes disrupted microglial activation in LPS-challenged mice [98]. Thus, IL-10/TGFβ signaling may be a key regulator of interactions between reactive astrocytes and microglia in neurodegenerative diseases.

### 5.6. Interactions between Reactive Glial Cells in Depression

The Nod-like receptor protein 3 (NLRP3) inflammasome has been reported to be an inducer of neuroinflammation that activates the secretion of inflammatory cytokine IL-1β in the mouse brain after fear stress [99]. The roles of the NLRP3 inflammasome in the pathogenesis of depression have also been reported recently [100,101], and astrocytes shifted to the neurotoxic state in depression. The microglia-specific ablation of NLRP3 was recently shown to markedly reduce A1-like astrocyte induction and lead to the alleviation of neuronal alteration both in vitro and in vivo depression-like mice models [102]. Furthermore, the authors in that study also found that the microglial NFkB pathway activated the NLRP3 inflammasome, which, in turn, activated the neuroinflammatory pathway led to the induction of neurotoxic astrocytes. Thus, since the NLRP3 inflammasome is one of the key players in the interactions between microglia and neurotoxic A1 astrocytes, it may serve as a target for the treatment of depression.

### 5.7. Interactions between Reactive Glial Cells in Phagocytic Activity

Microglia are known to exhibit phagocytic activity when they become activated in the CNS [103,104], and the metabotropic P2Y6 receptor expressed in microglia has been reported to be a trigger for their phagocytic activity [105]. Konishi et al. raised a question: “Which cells phagocytose debris in the brain when microglia’s activity is altered?” They and others had reported that sialic acid-binding immunoglobulin-like lectin H (Siglec-H) is expressed by the microglia in the brain but is not expressed in circulating monocytes and CNS-associated macrophages [106,107]. Surprisingly, Konishi et al. found that astrocytes become active, contacted, and engulfed microglial debris using their processes in Siglech^dtr^ mice, which have a high ablation ability with diphtheria toxin for microglia, in the hippocampal CA1 region [108]. Moreover, phagocytic activity from astrocytes instead of microglia has also been observed in interferon regulatory factor 8 (IRF8) knockout mice, which exhibit microglial dysfunction. IRF8 has already been reported to be a crucial transcriptional factor in microglial activation [109]. In addition, an RNA-seq analysis performed by Konishi et al. demonstrated that astrocytes express Tyro3/Axl/Mertk (TAM) phagocytic receptors even in the healthy brain. TAM receptors have already been reported to function as phagocytosis regulators in various tissues [110]. Thus, astrocytes have additional functions besides maintaining the physiological health of the brain.

## 6. Which Are Activated First, Astrocytes or Microglia?

The results of many studies have indicated that microglial activation is followed by astrocyte reactivation in several neurodegenerative diseases, including ALS [111], Huntington’s disease [112], neuropathic pain [113], and cortical TBI [114]. Shinozaki et al. reported finding that TBI-induced microglial activation followed by reactive astrocyte to a neuroprotective phenotype [115]. Interestingly, TBI has been found to induce primary microglial activation in the injury core and secondary neuroprotective astrocyte activation in the peri-injury region. Moreover, this interaction between active microglia and neuroprotective astrocytes has been shown to result from downregulation by microglia-derived cytokines of the P2Y_1_ purinergic receptor, which is essential for the acceleration of reactive astrogliosis. Regulating the level of expression of P2Y_1_ may provide a treatment for neurodegenerative diseases.

We have reported that 2-carba cyclic phosphatidic acid (2ccPA) suppresses inflammation via microglial polarization from the neurotoxic M1 phenotype to the neuroprotective M2 phenotype and is involved in the recovery from BBB breakdown in stab-wounded mouse cerebral cortex [116]. In addition, we have subsequently shown that 2ccPA suppresses neuronal apoptosis around stab wound injury sites in mouse brains and exerts a neuroprotective role via the extracellular matrix protein TN-C secreted by reactive astrocytes [40]. Therefore, we are now focusing on the functional signaling of the interaction between reactive astrocytes and reactive microglia through 2ccPA.

In the next section, we review the literature on the functional role of 2ccPA precisely because it plays neuroprotective roles in reactive astrocytes and microglia and may help in the development of treatments for neurological disorders.

## 7. Effect of Cyclic Phosphatidic Acid on Astrocyte Functions

### 7.1. Cyclic Phosphatidic Acid in the Brain

Cyclic phosphatidic acid (cPA) was first isolated from the myxoamoebae of the true slime mold *Physarum polycephalum* [117], and it has been found in a variety of mammalian tissues, including blood and brain [118,119,120]. cPA has a unique structure that contains a cyclic phosphate between *sn*-2 and *sn*-3 of the glycerol backbone. The fatty acid attached at the *sn*-1 position of the glycerol backbone varies. The most abundant species of fatty acid of cPA in human serum albumin, mouse plasma, mouse brain, and rat serum is palmitic acid (16:0) [118,119,120].

The cPA in mammalian blood is produced from lysophosphatidylcholine (LPC) by the lysophospholipase D (lysoPLD) activity of autotaxin (ATX) [121]. ATX catalyzes the formation of another bioactive lipid, lysophosphatidic acid (LPA) [122,123]. LPA has multiple effects on astrocytes via LPA receptors and modulates several neuronal functions [124,125]. Since cPA is a weak agonist of LPA receptors [126], cPA is also assumed to have multiple effects on astrocytes via LPA receptors. However, the bioactivity of cPA is often different from that of LPA. This is because the functions of LPA and cPA are dependent on cell type, receptor expression, specificity, affinity, and the signaling pathways stimulated. To understand the function of cPA, it is necessary to clarify not only the molecular mechanism of LPA receptor activation but also its production and degradation. Interestingly, one of the different activities of cPA and LPA is ATX inhibition [126]. cPA could be an endogenous inhibitor of ATX and may decrease LPA production. It has been reported that LPA was elevated in the cerebrospinal fluid of TBI patients and mice in a TBI model [127], and it was demonstrated that LPA plays an important role in tissue damage in a mouse model of TBI [127,128]. Furthermore, it was suggested that LPA stimulates tau hyperphosphorylation, leading to tau aggregation and the formation of paired helical filaments in AD brains [129]. Therefore, investigating the regulation of ATX expression and activity related to LPA levels in various physiological and pathological states of the brain is important. We are now interested in how ATX activity is regulated to generate LPA and the endogenous inhibitor of ATX, cPA, from LPC under different conditions. Moreover, in addition to ATX, several intracellular enzymes have been reported that catalyze the production of cPA [130,131]. One intracellular protein, phospholipase D2 (PLD2), catalyzes the formation of cPA from lysophosphatidylcholine (LPC) [130], and another is an intracellular membrane-bound enzyme, GDE7, that possesses lysophopholipase D activity, GDE7 and catalyzes the formation of cPA from lysophosphatidylserine as well as from LPC [131]. The cPA produced as a result of the action of these intracellular enzymes must be secreted extracellularly to inhibit ATX and/or stimulate LPA receptors. Although cPA is also known to be an antagonist of the peroxisome proliferator-activated receptorγ (PPARγ) [130], it is still unclear how intracellular cPA affects astrocytes.

Tissue cPA concentrations have been measured by liquid chromatography-tandem mass spectrometry (LC-MS/MS). The cPA concentration measured in the mouse brain was 10.99 ± 1.52 nmol/g-wet tissue [118] and almost the same as in the liver (9.88 ± 5.06) and kidney (7.53 ± 2.15), but much higher than in the heart (2.07 ± 0.88), lung (3.45 ± 1.65), and spleen (0.44 ± 0.19) [118]. Systemically administered cPA affects brain function within 20 s [132], indicating that cPA is able to pass through the BBB, although it remains unclear whether the cPA in the brain was produced in the brain or passed from circulating blood into the brain through the BBB. Based on the wet weight of the organs, brain tissue is the richest tissue in cPA.

### 7.2. Roles of Cyclic Phosphatidic Acid in the Brain

The functions of cyclic phosphatidic acid have been investigated by using in vivo animal models and in vitro. Several cPA analogs have been synthesized to investigate the functions of cPA [133,134]. 2ccPA is a chemically synthesized analog of cPA in which the *sn*-2 oxygen has been replaced by a methylene group, and it induces several biological activities much more potently than naturally occurring cPA does [133]. In this section, we describe the roles of cPA in the brain which are revealed by studies that used cPA and 2ccPA. A study conducted on an in vivo animal model revealed that cPA elicits neuroprotective actions for delay neuronal death following transient ischemia [135], following cuprizone-induced demyelination (demonstrated in a multiple sclerosis model) [136,137], and after a stab wound to the cerebral cortex [40,116]. In the rat model of transient ischemia, cPA and 2ccPA were found to protect hippocampal neurons and decrease glial fibrillary acidic protein (GFAP) immunoreactivity [135]. In the multiple sclerosis mouse model, cPA and 2ccPA suppressed demyelination and motor dysfunction, and 2ccPA decreased GFAP mRNA expression [136]. Interestingly, the amount of cPA in the brain was significantly decreased in the cuprizone-treated mice. In the mouse model of TBI, 2ccPA was found to suppress inflammatory cytokine expression and neuronal apoptosis after the injury [116], and 2ccPA increased the immunoreactivity of TN-C, an extracellular matrix protein in the vicinity of the wound region. 2ccPA also increased the mRNA expression levels of TN-C in primary cultured astrocytes, and the conditioned medium of 2ccPA-treated astrocytes suppressed cortical neuron apoptosis [40]. These results indicated that cPA has a neuroprotective function in the brain that is exerted via astrocyte responses.

### 7.3. Role of Cyclic Phosphatidic Acid in Astrocytes

Thus far, six types of LPA receptors have been cloned and characterized. They belong to two families: LPA1-3 receptors are members of the endothelial differentiation gene (EDG) family and LPA4-6 receptors are members of the purinergic receptor family, both families are G-protein coupled receptors (GPCR) [138,139]. Some of the biological functions of cPA have been shown to be induced by the activation of LPA receptors. The ligand specificity of LPA receptors may be responsible for the specific biological functions of cPA. LPA1 and LPA6 are more highly expressed in astrocytes than other LPARs are [140]. Both cPA and 2ccPA have LPA1 agonist activity [126], whereas 2ccPA has LPA6 agonist activity, but cPA does not. Thus, cPA can function via LPA1 alone and not via LPA6, whereas 2ccPA can function via both LPA1 and LPA6. Interestingly, our recent study showed that 2ccPA increased cell number and cell survival during oxidative stress via mainly the LPA1/3 receptors in astrocytes [140]

cPA is an amphiphilic molecule and can stimulate PPAR-γ, a nuclear receptor family protein [130]. Since both neuroprotective and anti-inflammatory properties of PPAR-γ agonists have been reported [141], cPA can bind to PPAR-γ and inhibit pro-inflammatory mediator expression and upregulate anti-inflammatory mediator expression.

Moreover, adenine nucleotide translocase 2 (ANT2) has recently been found to be a putative target protein for 2ccPA in microglial cells [138], and binding of 2ccPA to ANT2 has been shown to protect microglial cells from apoptosis [142]. cPA may function by binding to ANT2 in astrocytes. In conclusion, cPA and 2ccPA affect astrocytes, resulting in it has neuroprotective functions in animal models. It is unclear whether the decrease in GFAP caused by cPA is a consequence or cause of pathological conditions. More precise experiments will need to be conducted in order to understand the effect of cPA on astrocyte responses. Our hypothesis to explain the effect of cPA on the astrocyte immune response is shown in Figure 1.

## 8. Conclusions

We have proposed the involvement of several factors in the interactions between reactive astrocytes and reactive microglia in various CNS diseases and in BBB integrity. The novel molecules cPA and 2ccPA may contribute to the treatment of neurodegenerative disorders.

## Figures and Tables

**Figure 1 ijms-24-01599-f001:**
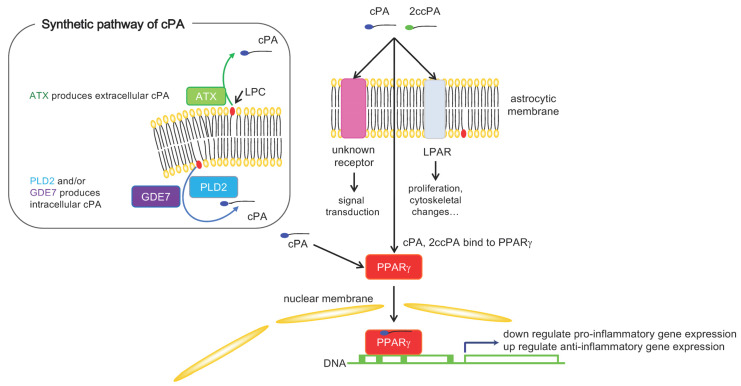
Hypothesis to explain the effect of cPA on the astrocyte immune response. cPA in the brain is produced by intracellular or extracellular enzymes [121,130,131]. Extracellular cPA stimulates six different LPARs [126] and induces cellular responses, including cell proliferation and cytoskeletal changes [140]. The intracellular and/or extracellular cPA bind PPAR-γ and induce anti-inflammatory responses [130,141]. cPA also binds ANT2, and this binding is related to protecting cells from apoptotic death [142]. There may be unknown cPA-specific receptors. (This figure adapted from Ref. [143]).

## Data Availability

Not applicable.

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
