# Peer review of "Astrocytic Neuroimmunological Roles Interacting with Microglial Cells in Neurodegenerative Diseases"

_ijms, 2023, doi:10.3390/ijms24021599_

Round 1

Reviewer 1 Report

This review article focuses on the astrocytic function in the extracellular matrix and the interaction between reactive astrocytes and reactive microglia under physiological and pathological conditions. The potential interaction of astrocytes with the basement membrane seems interesting, and part of the description related to Tenascin-C (TN-C) is well done.

The authors wrote: "Astrocytes, as well as microglia, express phagocytotic activity on neuronal debris in damaged tissue…". For clarity of the article, the section titled "Phagocytic activity of reactive astrocytes and their signaling" should be written after the brief introduction to microglia and connected to microglial phagocytosis. It may be interesting to highlight the phagocytic role of astrocytes in physiological and pathological conditions, especially since they participate in the elimination of synapses and neuronal debris in the developing and injured brain.

What is the full name of CD49f?

The last part of the review article describes the role of cyclic phosphatidic acid (cPA), the naturally occurring analog of LPA, and its synthetic analogs. Since autotaxin (ATX) is involved in LPA and cPA formation, the authors should mention that ATX is subject to feedback inhibition by lysophosphatidic acid (LPA). Moreover, cPA and its analogs may inhibit ATX-mediated LPA production. How is ATX regulated to generate LPA/cPA under different physiological or pathological conditions? For example, increased ATX expression is detected in activated astrocytes after brain injury and in the frontal cortex of patients with Alzheimer's type dementia.

The figures were not attached to the manuscript.

Author Response

Reply to comments from Reviewer 1:

Thank you very much for your valuable advice to our review paper.

Revised points are colored in yellow in our revised manuscript and replied point by point as below.

Comment 1:

The authors wrote: "Astrocytes, as well as microglia, express phagocytotic activity on neuronal debris in damaged tissue…". For clarity of the article, the section titled "Phagocytic activity of reactive astrocytes and their signaling" should be written after the brief introduction to microglia and connected to microglial phagocytosis. It may be interesting to highlight the phagocytic role of

astrocytes in physiological and pathological conditions, especially since they participate in the elimination of synapses and neuronal debris in the developing and injured brain.

Reply from the authors to Comment 1:

According to the reviewer’s important suggestion, we changed the order of the sections as “3. What are microglia?” in line 203-250, page 10-13 and then, “4. Phagocytic activity of reactive astrocytes and their signaling” in line 253-289, page 13-14.

Like the reviewer’s recommendation, we think these changes must be attracted reader’s interest.

Comment 2:

What is the full name of CD49f?

Reply from the authors to Comment 2:

Thank you for the important point.

In fact, CD49f is encoded by integrin alpha 6 gene.

Thus, we amended the sentence from “CD49f has been reported…” in our original manuscript to “CD49f, encoded by integrin alpha 6 gene, has been reported…” in line 92, page 5 in our revised manuscript.

Comment 3:

The last part of the review article describes the role of cyclic phosphatidic acid (cPA), the naturally occurring analog of LPA, and its synthetic analogs. Since autotaxin (ATX) is involved in LPA and cPA formation, the authors should mention that ATX is subject to feedback inhibition by lysophosphatidic acid (LPA). Moreover, cPA and its analogs may inhibit ATX-mediated LPA

production. How is ATX regulated to generate LPA/cPA under different physiological or pathological conditions? For example, increased ATX expression is detected in activated astrocytes after brain injury and in the frontal cortex of patients with Alzheimer's type dementia.

Reply from the authors to Comment 3:

Since the reviewer gave us very critical advice for the section 7, we inserted some of the sentences with additional references to clarify the section as below.

Some words “and modulates several neuronal functions” are added in line 437 with one reference [125] and “is a weak agonist of” instead of “activates” in line 438, page 21 in revised manuscript.

  1. de Sampaio e Spohr, T.C.L.; Dezonne, R.S.; Rehen, S.K.; Alcantara Gomes, F.C. LPA-Primed Astrocytes Induce Axonal Outgrowth of Cortical Progenitors by Activating PKA Signaling Pathways and Modulating Extracellular Matrix Proteins. Front Cell Neurosci 2014, 8, 1–9, doi:10.3389/fncel.2014.00296.

Furthermore, we inserted the sentences with three additional references instead of “Several intracellular enzymes that catalyze the formation of cPA have been reported” in line 439-455, page 21-22 in revised manuscript.

“However, the bioactivity of cPA is often different from that of LPA. This is because the functions of LPA and cPA are dependent on cell type, receptor expression, specificity, affinity, and the signaling pathways stimulated. To understand the function of cPA, it is necessary to clarify not only the molecular mechanism of LPA receptor activation, but also its production and degradation. Interestingly, one of the different activities of cPA and LPA is ATX inhibition [126]. cPA could be an endogenous inhibitor of ATX and may decrease LPA production. It has been reported that LPA was elevated in the cerebrospinal fluid of TBI patients and of mice in a TBI model [127]. And it was demonstrated that LPA plays an important role in tissue damage in a mouse model of TBI [127,128]. Furthermore, it was suggested that LPA stimulates tau hyperphosphorylation, leading to tau aggregation and the formation of paired helical filaments in AD brains [129]. Therefore, to investigate the regulation of ATX expression and activity related to LPA levels in various physiological and pathological states of the brain is important. We are now interested in how ATX activity is regulated to generate LPA and the endogenous inhibitor of ATX, cPA, from LPC under different conditions. Moreover, in addition to ATX, several intracellular enzymes have been reported that catalyze the production of cPA.”

  1. Crack, P.J.; Zhang, M.; Morganti-Kossmann, M.C.; Morris, A.J.; Wojciak, J.M.; Fleming, J.K.; Karve, I.; Wright, D.; Sashindranath, M.; Goldshmit, Y.; et al. Anti-Lysophosphatidic Acid Antibodies Improve Traumatic Brain Injury Outcomes. J Neuroinflammation 2014, 11, doi:10.1186/1742-2094-11-37.
  2. Goldshmit, Y.; Matteo, R.; Sztal, T.; Ellett, F.; Frisca, F.; Moreno, K.; Crombie, D.; Lieschke, G.J.; Currie, P.D.; Sabbadini, R.A.; et al. Blockage of Lysophosphatidic Acid Signaling Improves Spinal Cord Injury Outcomes. American Journal of Pathology 2012, 181, 978–992, doi:10.1016/j.ajpath.2012.06.007.
  3. Ramesh, S.; Govindarajulu, M.; Suppiramaniam, V.; Moore, T.; Dhanasekaran, M. Autotaxin-Lysophosphatidic Acid Signaling in Alzheimer’s Disease. Int J Mol Sci 2018, 19, doi:10.3390/ijms19061827.

Comment 4:

The figures were not attached to the manuscript.

Reply from the authors to Comment 4:

We asked Mr. Jesse Yi, who is in charge in MDPI, to send our figure 1 and 2 to the reviewers to show the other day.

For confirmation, please find attached file of our figure 1 and 2.

Reviewer 2 Report

This is an enjoyable article to read. This is a nice summary of literature related to interactions between reactive astrocytes and reactive microglia in normal states and in neurodegenerative diseases.  Just as the authors stated as a reason for this review. This manuscript can serve as a foundation in obtaining and overview as well as opening ideas of areas of interest to continue investigations.
I only have a few suggestions:

1.    The figures did not appear on the PDF file I downloaded. Can these be sent to view ?
2.    Abbreviations:  LPS; lipopolysaccharide;
But in
Section 7.1 “……cPA from lysophosphatidylserine (LPS) as well as from LPC [127]. “

So LPS is defined in two different ways in the paper.

Author Response

Reply to comments from Reviewer 2:

Thank you very much for your valuable suggestions to our review paper.

We revised point by point as below.

Comment 1:

  1. The figures did not appear on the PDF file I downloaded. Can these be sent to view ?

Reply from the authors to Comment 1:

We asked Mr. Jesse Yi, who is in charge in MDPI, to send our figure 1 and 2 to the reviewers to show the other day.

For confirmation, please find attached file of our figure 1 and 2.

Comment 2:

  1. Abbreviations: LPS; lipopolysaccharide;

But in Section 7.1 “……cPA from lysophosphatidylserine (LPS) as well as from LPC [127]. “

So LPS is defined in two different ways in the paper.

Reply from the authors to Comment 2:

To avoid readers’ confusion, we deleted “(LPS)” from our revised manuscript in line 459, page 22.

Round 2

Reviewer 1 Report

The authors have adequately addressed to my comments in the
revised version of the manuscript. Therefore, I have no further comments.

Reviewer 2 Report

Thank you for the edits and the figures.

The figures are very nicely produced. They help in the overview.